Controller placement with critical switch aware in software-defined network (CPCSA)

Muhammed Yusuf Nura 1 2 ymnura@atbu.edu.ng
Abu Bakar Kamalrulnizam 1
http://orcid.org/0000-0002-3820-3378 Isyaku Babangida 1 3 bangis4u@gmail.com
http://orcid.org/0000-0001-7474-9100 Abdelmaboud Abdelzahir 4
Nagmeldin Wamda 5
1 Department of Computer Science, Faculty of Computing, Universiti Teknologi Malaysia , Johor, Johor Bahru , Malaysia
2 Department of Mathematical Science, Faculty of Sciences, Abubakar Tafawa Balewa University , Bauchi , Nigeria
3 Department of Computer Science, Faculty of Computing and Information Technology, Sule Lamido University , Kafin Hausa, Jigawa State , Nigeria
4 Department of Information Systems, King Khalid University , Abha , Saudi Arabia
5 Department of Information Systems, College of Computer Engineering and Sciences, Prince Sattam bin Abdulaziz University , Al-Kharj , Saudi Arabia
Sadek Rowayda
Electronic publication date: 2023 Dec 19
Publication date: 2023
Volume: 9
Electronic Location ID: e1698
Received 2023 Aug 18; Accepted 2023 Oct 24
Copyright: © 2023 Muhammed Yusuf et al.
Copyright year: 2023
Copyright holder: Muhammed Yusuf et al.
License: This is an open access article distributed under the terms of the Creative Commons Attribution License, which permits unrestricted use, distribution, reproduction and adaptation in any medium and for any purpose provided that it is properly attributed. For attribution, the original author(s), title, publication source (PeerJ Computer Science) and either DOI or URL of the article must be cited.
License URL: https://creativecommons.org/licenses/by/4.0/

Keywords: SDN, Controller placement, Controller overhead, Switch role, Network partition

Funding: Deanship of Scientific Research at King Khalid University through large group Research Project RGP.2/175/44 This work is funded by Deanship of Scientific Research at King Khalid University through the large group Research Project under grant number (RGP.2/175/44). The funders had no role in study design, data collection and analysis, decision to publish, or preparation of the manuscript.

==============================
Software-defined networking (SDN) is a networking architecture with improved efficiency achieved by moving networking decisions from the data plane to provide them critically at the control plane. In a traditional SDN, typically, a single controller is used. However, the complexity of modern networks due to their size and high traffic volume with varied quality of service requirements have introduced high control message communications overhead on the controller. Similarly, the solution found using multiple distributed controllers brings forth the ‘controller placement problem’ (CPP). Incorporating switch roles in the CPP modelling during network partitioning for controller placement has not been adequately considered by any existing CPP techniques. This article proposes the controller placement algorithm with network partition based on critical switch awareness (CPCSA). CPCSA identifies critical switch in the software defined wide area network (SDWAN) and then partition the network based on the criticality. Subsequently, a controller is assigned to each partition to improve control messages communication overhead, loss, throughput, and flow setup delay. The CPSCSA experimented with real network topologies obtained from the Internet Topology Zoo. Results show that CPCSA has achieved an aggregate reduction in the controller’s overhead by 73%, loss by 51%, and latency by 16% while improving throughput by 16% compared to the benchmark algorithms.

Introduction

Software-defined networking (SDN) is an emerging network paradigm offering simple network management by separating network control logic and data forwarding elements. This way, the control plane (CP) is responsible for providing and enforcing network policies on the switches at the data plane (DP). To achieve this, the controller uses a link layer discovery protocol (LLDP) to identify the OpenFlow switches connected at the DP (Yusuf et al., 2023a). It then continuously monitors them for changes due to events like failures or the arrival of new flows. It collects network statistics concerning traffic arrival patterns, traffic types, and other changes for various applications like routing, congestion control, and security to run their algorithm instances (Isyaku & Bakar, 2023). For any state change at DP, the controller must immediately recalculate updated instructions for the DP switches, sending them as a packet-out message to all edge switches (for ARP) and a flow-mod message to all switches along the same path for installation on their flow tables (Yusuf et al., 2023c). Recently, the controller has been experiencing a substantial increase in communication overhead due to an exponential growth in new flow arrival rates caused by the proliferation of Internet of Things (IoT) devices and the expansion of network size (Firouz et al., 2021). Consequently, the DP may frequently encounter state change events like link failure (Isyaku et al., 2023), requiring the controller to reconfigure new rules (Isyaku et al., 2021).

This process has implications for the workload of the controller. For instance, if a flow traverses an average path length of six switches and the network has 100 edge switches, the controller is estimated to spend around 6 ms to handle each flow (Zhao et al., 2017a). A prior study reports that processing these messages adds an overhead and delay of approximately 0.5 and 0.2 ms, respectively. As a result, the cumulative burden on the controller amounts to (0.5 * 6 + 0.2 * 100) (Zhao et al., 2017a). Moreover, another study highlights a direct correlation between the number of switches in a network and the volume of flow setup requests. According to Curtis et al. (2011), configuring a flow route for a network with N switches incurs an overall cost of approximately 94 + 144N, with an additional 88N byte attributed to flow-removed messages. Thus, CP design is critical to the performance of SDN.

A single controller (csCP) design is widely used for small network sizes. However, it may fail to give the desired performance due to high control message processing overhead. It also exhibits reliability concerns due to a single failure point (SPOF), as the failure tendencies are higher when the network is large. As such researchers leverage multiple controllers (dmCP), which better performance compared to csCP. Figure 1 illustrates the differences between the former and the latter. For example, an extensive network may have switches that can generate up to 750 to 20,000 flow per second (Isyaku et al., 2021); others say it might reach up to 10 million flow requests per second (Ahmad & Mir, 2021; Yan et al., 2021). Unfortunately, this is beyond the capacity of a single controller, as some controllers can only accommodate 6,000 flow requests per second (Hu et al., 2017). On the other hand, designing the CP with multiple controllers opens up a controller placement problem (CPP) challenge. For any given network, the CPP deals with finding and optimising (i) the number of controllers in the network. (ii) The controllers should be placed strategically on the network to minimise congestion, overhead, and Latency between controllers and switches. Heller, Sherwood & Mckeown (2012), who initiated the concept of (CPP), built their solution while considering the impact of Latency. The solution performs well for small-scale networks; however, it ignores the effects of scalability, reliability, and congestion in large networks such as WAN. Assigning controllers to switches in an extensive network can exhibit an imbalance distribution of load among the controllers. Therefore, for a software-defined wide area network (SDWAN), a partitioning algorithm is employed to cluster the network into smaller subnets for controller placement (Killi & Rao, 2019).

Figure 1 Control plane architecture.

Single control plane architecture (A) and multiple controllers (B).

Several CPP solutions employ network partitioning techniques in their approaches. For example, methods such as Killi, Reddy & Rao (2019), Kuang et al. (2018), Liu, Liu & Xie (2016), Xiao et al. (2014), Wang et al. (2018), Xiao et al. (2016), Yang et al. (2019a, 2019b) and Zhu et al. (2017) are designed based on k-means. A K-median is used by Liu et al. (2018) and Kobo, Abu-Mahfouz & Hancke (2019), while Xiao et al. (2014, 2016), Aoki & Shinomiya (2015, 2016), Zhao & Wu (2017) used Spectral Clustering. Density-based Clustering, Affinity Propagation, and Partitioning Around Medoids (PAM) are also used in Liao et al. (2017), Zhao et al. (2017b), Bannour, Souihi & Mellouk (2017) and Dvir, Haddad & Zilberman (2018). Others hybridised two techniques in their solution (Yang et al., 2019a, 2019b; Firouz et al., 2021; Manoharan, 2021). All these techniques share the common idea of partitioning the SDWAN into smaller sub-domains, allowing for assigning one or more exclusive controllers to cover each subdomain. The k-means algorithm is one of the common methodologies used to partition a network topology. It uses Euclidean distance as its similarity metric during the partition process. However, computing Euclidean distance in real networks is not always possible due to the lack of physically connected pathways in some instances. Similarly, the strategy has no generally agreed-upon way to determine the first k partitions. The method varies in how it initialises the first set of cluster heads. Hence, the initial cluster head selection significantly affects the solution quality; thus, it is a significant limitation.

On the other hand, PAM is quite similar to k-means, except that it minimises the impact of outliers by selecting a node at the cluster’s centre as the head. Although PAM does not require prior knowledge of k, it has a considerably high complexity to the tune of about cubic time. Additionally, while these approaches may be suitable for initial controller placement, repeatedly segmenting the entire network to adapt to its dynamic nature is unrealistic. At the same time, spectral clustering tends to produce small, isolated components and clusters of skewed sizes. In addition, all the solutions did not quantify the controller’s overhead and response time (RT) in their performance validation.

In the rapidly evolving landscape of SDN, the efficient placement of controllers plays a pivotal role in network performance and reliability. This article addresses this critical challenge by introducing an innovative approach that optimises controller placement and considers the impact on critical switches within the network. The existing solution did not adequately consider the roles of switches in the network. It is important to note that switches have different roles; some switchers are very critical, and others are non-critical. The former can have a significant impact on the efficient controller placement solution. Identifying critical switches is crucial for optimal controller placement during network partitioning decisions. Critical switches possess a high degree and betweenness criticality measures that tend to send higher flow rule requests to the controller. As a result, they often augment the flow setup delay and cause high update operations. This problem results in additional overhead on the controller if multiple critical switches reside in the same partition. Therefore, this article proposes the Controller placement algorithm with network partition based on critical switch awareness (CPCSA) to mitigate these issues. CPCSA identifies critical switch in the SDWAN and then partition the network based on the criticality. Subsequently, a controller is assigned to each partition to improve control messages communication Overhead and other dependent QoS metrics like loss, throughput, and flow setup delay. We itemized the contributions of this article as follows. We devised a network partitioning model based on the switch role in the network to determine the number of controllers.

A switch to controller placement strategy was introduced based on switch criticality factor to improve the control plane’s performance.

The performance evaluation result of CPCSA using real networks from Internet Topology Zoo in comparison to other relevant CPP algorithms.

The remainder of the article is structured as follows: ‘Related Works’ related works in SDN. ‘Materials and Method’ analyses the problem. Next, Section 4 presents the proposed solution. Then, ‘Results’ describes the experimental setup and performance evaluation. Lastly, ‘Conclusion’ concludes the study and makes recommendations for future research.

Related works

Selecting a suitable position in SDWAN for controller placement is crucial to its performance (Heller, Sherwood & Mckeown, 2012). Inappropriate controller placement can increase communication overhead and flow setup delay. Therefore, several CPP solutions have been proposed (Yusuf et al., 2023b). The CPP solutions presented in Xiao et al. (2014, 2016), Aoki & Shinomiya (2015, 2016) and Zhao & Wu (2017) utilised spectral Clustering to partition the wide-area Network into many subnetworks. Some authors infer the count of subnets by exploiting the concept of eigenvectors, using the Haversine equation to calculate the similarity graph. Each resulting subnetwork is assigned a dedicated controller at a location that minimises the control message latency. Researchers in Zhao & Wu (2017) formulate the CPP as an integer linear programming (ILP) with the optimisation objective of reducing the network cost. They design a heuristic method to solve the ILP. However, spectral clustering tends to produce small, isolated components and clusters of similar sizes. In addition, all the solutions did not quantify the controller overhead and response time (RT) in the performance validation.

In a different approach to formulating a clustering-based CPP (Zhu, Chai & Chen, 2017), researchers utilise integer programming (IP). They reduce the network’s transmission time by employing a modified version of k-means with the shortest path as the similarity metric. In Zhao et al. (2017b), the authors formulate a binary variable model of the CPP and cluster it using an affinity propagation technique (APT). APT maximised similarity across short distances and moderated preference control to a mean value. In another approach, Liao et al. (2017) propose density-based controller placement (DBCP) to partition a network into various sub-networks. The DBCP grouped tightly connected switches within the same subnet and less-connected switches in a different subnet. The value of k and members of each subnet is determined based on the distance to a higher-density node. Each sub-network is assigned a single controller. In other techniques, PAM-B clustering and NSGA-II were utilised by Bannour, Souihi & Mellouk (2017) to solve the Network partitioned-based CPP with the multi-objective problem of optimising Latency, capacity, and availability. In another approach, using the shortest path as the similarity metric (Wang et al., 2018, 2016), partitioned a network for CPP using k-means. Starting with a random centroid, the algorithm iterates continuously until it divides the network into k clusters. In a similar effort, researchers utilised simulated annealing (SA) and the k-median algorithm (Liu et al., 2018) to determine the optimal location for a satellite gateway in a 5G network, aiming to reduce latency. The authors implemented a clustering strategy to improve connectivity reliability between satellites and controller nodes. Also, Kuang et al. (2018) confronts the network partitioning problem by employing the k*-means for a CPP. Initialised the partitioning with more than k clusters and later merged the nodes into the k clusters recursively based on the shortest path distance and cluster load. While in a different approach proposed by Killi, Reddy & Rao (2019), for Network partition-based controller placement to reduce latency, the authors utilise a k-means algorithm with initialisation based on cooperative game theory. Cooperative game with a set of switches as players are used to mimicking the division of the Network into subnetworks. The switches attempt to build alliances with other switches to increase their value. They also suggest two variations of the cooperative k-means technique to create size-balanced partitions. However, these approaches did not consider load balance issues. Dvir, Haddad & Zilberman (2018) formulated the CPP as an IP. The network was divided into partitions using a k-medoid clustering technique. However, the value of k is determined via a brute-force approach. In contrast, CPP was tackled using a k-centre/k-median clustering strategy by Kobo, Abu-Mahfouz & Hancke (2019). The authors suggested creating a local and global controller hierarchy. When a controller fails, it is replaced using the re-election procedure. To assess load balancing (Yang et al., 2019a, 2019b) defines two distinct cost functions regarding the network topology structure and flow traffic distribution. They then hybridise the network partition scheme to tackle the problem of where to locate the load-balancing controller. Each of the numerous sub-domains that comprise the overall Network has one dedicated controller. Finally, a simulated annealing partition-based k-means (SAPKM) to address the placement is proposed. SAPKM incorporates a centroid-based clustering to achieve load-balancing among the controllers. The k-means algorithm uses Euclidean distance as its similarity metric. However, the problem is that it is not always possible to compute the Euclidean distance in real networks due to the lack of physically connected pathways. Similarly, k-means has no agreed-upon way to determine the first k partitions. The method varies in how it initialises the first set of clusters head. Thus, the initial cluster head selection significantly affects the solution quality in k-means; this is considered a significant limitation. On the other hand, PAM is quite like k-means, except that it establishes a node in the cluster’s centre as the head to minimise the effects of the outliers. Although they do not require prior knowledge of k, they have a significantly higher level of complexity to the tune of about cubic time. At the same time, spectral clustering tends to produce small, isolated components and clusters of similar sizes.

Network clustering for CPP using data field theory (DFT) was proposed by Li et al. (2019). The DFT considers the strength of the wireless nodes’ transmissions and reception signal power to determine the controller placement inside each cluster to reduce Latency and energy. While Ali & Roh (2022) and Ali, Lee & Roh (2019) presents an SDN partition strategy for controller placement in IoT environments to reduce latency using the analytical network process (ANP). The authors thoughtfully consider multiple latency-inducing parameters to guide their ranking and selection process with ANP. However, it’s worth noting that one parameter that wasn’t considered in their analysis is the controller’s overhead. This omission is significant as it can impact performance and should ideally be factored into such an optimization strategy.

Another work (Manoharan, 2021) employed a graph theory to identify the number of controllers and their initial location. A Depth-First-Search algorithm is applied to determine Articulation Points (AP) based on two conditions. To obtain the required number of controllers and placement positions, they utilize APs. Additionally, they discretize a supervised machine learning concept using Manta-Ray Foraging Optimization (MRFO) and Salp Swarm Algorithm (SSA) to solve CPP based on network partitioning (Firouz et al., 2021). However, the lack of a standardized and rich dataset for model training has been a serious concern in any AI-based solution for SDN problems (Isyaku et al., 2020; Elsayed et al., 2019). However, privacy and confidentiality issues associated with Networks have made sharing this data difficult and scarce. Additionally, the approaches may be suitable for acquiring the first controller placement. However, it is unrealistic to repeatedly segment the entire Network to meet the evolution of dynamic network changes. Thus, they lack an adaptable CPP that responds to the dynamics of each given network. Therefore, based on the discussed literature, it can be conclude that all the solutions have not adequetely consider the switch's role in the Network to identify and separate a set of critical from non-critical switches. Recognizing the critical switches is crucial during network partition decisions for optimum controller placement. Such sets of switches possess high degree and betweenness criticality measures with many rules in their flow table entries. As a result, they often augment the flow setup delay and cause more update operations. The problem leads to additional overhead on the controller if multiple critical switches are in the same partition. See Table 1 for the summary of these approaches.

Table 1 Network partitioned-based CPP.

Article	Problem formulation	Partition/solution approach	Network topology properties	Performance metrics considered	Weakness	
Path	Switch role	Metrics	Latency	Overhead	Loss	Partition approach	Performance metrics	
Wang et al. (2016)	MILP	Heuristics	✓	X	X	✓	✓	X	Not partitioned	Throughput and loss unaccounted	
Killi & Rao (2019), Kuang et al. (2018)	Network partitioning	Spectral clustering	✓	X	Eigen
vectors	✓	X	X	Tend to produce small, isolated components and clusters with similar sizes	High CP overhead, Poor load balancing &
CP overhead and throughput	
Yang et al. (2019b), Zhu et al. (2017)	✓	X	✓	X	X	
Li et al. (2019)	Node Burden	✓	✓	Traversal set	✓	X	X	
Liu et al. (2018)	ILP	Spectral clustering	✓	X	Eigen
vectors	✓	X	X	
Bannour, Souihi & Mellouk (2017)	K-Means	✓	X	Euclidean distance	✓	X	X	Random centre initialisation stage, the number of cluster determinations	
Modified-AP (Aoki & Shinomiya, 2015)	BIP	Affinity propagation	✓	X	Shortest distance	✓	X	X	Not partitioned	
Kobo, Abu-Mahfouz & Hancke (2019)	Network partitioning	Density-based clustering	✓	✓	Density	✓	X	X	NA	
Aoki & Shinomiya (2016)	MOCO	PAM-B	✓	X	Dijkstra	✓	X	X	Quadratic running time complexity	
SACA (Xiao et al., 2016)	Mathematical	K-Median, SA	✓	X	Euclidean distance	✓	X	X	Random centre initialisation, number of cluster determinations, the use of “means” limit its expression level, Euclidean distance might not get a path physically connected path, one size fits it-all effect, outliers, and noise	
Hu et al. (2017)	Network partitioning	K-Means	✓	X	✓	X	X	
Zhao & Wu (2017)	IP	K-Mediod		X	✓	X	X	
Killi, Reddy & Rao (2019), Dvir, Haddad & Zilberman (2018)	Mathematical model	K-Means	✓	X	✓	X	X	
Ali & Roh (2022)	Clique-based	✓	X	Shortest distance	✓	X	X	Too rigid to use in practice. It tends to produce maximally cohesive subgraph	The clique property cant guarantee optimum RT	
SACKM (Liu, Liu & Xie, 2016; Xiao et al., 2014)	Hybridised SA
with K-Means	✓	X	Euclidean distance	✓	X	X	K-means limitation, SA limited memory to track tested solutions, low improvement rate,	Ignore the
CP overhead, LB, and throughput	
Manoharan (2021)	Data field theory	X	X	Signal strength	✓	X	X	Interference	
Yang et al. (2019a)	IP	K-Median	✓	X	Haversine	✓	X	X	Random centre initialisation stage, the number of cluster determinations,	
Yan et al. (2021)	Mathematical model	K-means with game theory	✓	X	Euclidean distance	✓	X	X	
PHCPA (Yusuf et al., 2023a)	AI	MRFO with Salp Swarm	✓	X	Cosine Haversine	✓	X	X	Lack of sufficient training dataset	Increased PPT, control message overhead	
PITS (Liao et al., 2017)	Graph theory,	DFS	--	---	----	✓	X	X	
GravCPA (Ali, Lee & Roh, 2019)	LP	Louvain algorithms	X	Node Traffic	Euclidean	✓	X	X	LPA and gravitation are vulnerable to oscillations and non-unique results	
ECP (Isyaku et al., 2020)	MILP	Linearization & Supermodular	X	X	----	✓	✓	X	The CP overhead will likely resurface due to not partitioning the network into smaller clusters.	
Elsayed et al. (2019)	Greedy	None	X	X	X	✓	✓	X	Network properties not considered	No controller placement module	
Note:

PITS, Pareto integrated Tabu search; SA, simulated annealing.

Materials and Methods

Analysis of controller overhead

SDN controller overhead refers to the computational and resource requirements imposed on the SDN controller as it manages and controls the network. Although, the controller operates based on either proactive or reactive mode. The former may have lower overhead but may not cope with the real network []. The latter is widely used due to its flexibility in real-time network. However, any newly arrived Flow nFi at switch si∈S without corresponding forwarding rule entries in its flow table will introduce an overhead of composing and sending a Packet_IN message to its controller SProverhead on the switch. Likewise, on its part, the controller C also suffers the overhead of computing the required forwarding rule and subsequent installation in the switches si∈S flow Table via Packet_OUT message CProverhead. Due to these overheads, the new flow nFi, will experience a path setup time delay FSetUpSC, while waiting to be directed by a controller C. The flow/path setup delay emanates from five sources (i) a queue waiting time wtS at the switch Si before being served for duration stS, (ii) a switch si to controller C Packet_IN message propagation time Pin(si,C) (iii) a queue waiting time wtC at controller C before being served for (iv) a duration stC and (v) controller C to switch S Packet_OUT message propagation time Pout(C,Si). Therefore, cumulatively, the flow setup time delay is determined by.

(1) FSetUp=wtS+stS+Pin(Si,C)+wtC+stC+Pout(C,Si)

Equation (1) above fundamentally comprised the switch Si processing overhead, the controller C processing overhead, and the round-trip time between switch Si,and the controller C, given by Eqs. (2)–(4), respectively.

(2) SiProverhead=wtS+stS

(3) CProverhead=wtC+stC

(4) RTT=Pin(S,C)+Pout(C,S)

Considering a network topology with anS set of switches si∈S and E, as the communication links between the switches, can be represented as graph G=(S,E). Any mapping of a set of switches si∈S with a controller C impose an overhead CProverhead on the controller that is directly proportional to the cost of the flow rule setup request and subsequent rule installation in the flow table.

(5) CProverhead∝∑SProverhead

The SProverhead at the switch Si is determined by the load of the switch due to the new flow nFi arrival rate from both the external source ( Host) and internal source ( sj). As stated in Eq. (5), the overhead SProverhead directly increases the CProverhead. Therefore, if nFh0,Si, denote the external new flows arrival rate at the switch si from host h0. Let Xim∈{0,1} variables indicate whether the switch si is under the control of the controller Cm or not, using Xia={1, if si →Cm0, if si ↛Cm. Thus, the nFi arrival rate at Si from host h0 will induce rule computation overhead on the controller equivalent to:

(6) ∑si∈S⁡(nFh0,Si)Xim

Hence if nFh0,Si, denote the internal new flows arrival rate at the OpenFlow switch Sj from host Si. The arrival rate will induce rule computation overhead at the SDN controller Cm equals to

(7) ∑si∈S⁡(nFSi,Sj)Xim

Therefore, for all the OpenFlow switches controlled by the controller Cm, The total overall overhead on the controller for rules installation in the OpenFlow switch Si is equal to:

(8) CProverhead=∑si∈S⁡(nFh0,Si)Xim+∑sisj∈S⁡(nFSi,Sj)Xim+∑sisj∈S⁡(nFSi,Sj)Xim+∑si∈S⁡(nFSi,h0,)Xim

The objective is to minimize the CProverhead to improve the overall FSetUp and other QoS metrics. High controller overhead directly increases flow setup time which consequently causes performance retardation, especially for traffic with deadline violation constraints.

Design of the proposed solution

The proposed controller placement algorithm with critical switch awareness (CPCSA) for software-defined wide area network partitioned the network based on the switch role and assigned the required number of controllers to each partition. The operational procedure of CPCSA consists of three phases, with the output of each phase serving as input to the next phase. (i) The critical switch identification phase (CSIP) for reading the network topology to identify critical switches. (ii) Network partition phase (NPP) for partitioning the discovered topology based on the number of critical switches identified in (CSIP) and (iii) controller placement and assignment phase (CPAP), which uses the mathematical concept of facility location method to select a strategic position to place an SDN controller for each of the partitions formed in (NPP). This way, CPCSA placed an SDN controller in each partition formed based on the distance between the critical and non-critical switches within the partition to minimize the communication overhead and delay. ‘Network topology read phase’, ‘Switch role and critical switch identification phase (CSIP)’, ‘Network partition based on switch criticality’ and ‘Critical switch aware controller placement (CSACP)’ provide a detailed description of each phase. At the same time, the flowchart shown in Fig. 2 presents the overall procedure of the proposed algorithm (CPCSA).

Figure 2 CPCSA flow chart.

Network model and placement metrics

Consider an SDWAN topology modelled as a graph G=(V,E), with V representing a set of nodes and E the communication links between the nodes. The network node V comprised a group of OpenFlow switches S and an SDN Controllers C, i.e., i.e.,S,C∈V. The collection of the OpenFlow Switches S includes critical switches (CS) and non-critical switches (nCS). For controller placement, the technique partitions G into multiple sub-nets SDWAN_Partitionsi to improve latency performance and reduce a Controller’s overhead. In this study, we formulate the network partition problem by considering the switch’s role in the Network. This help in identifying the critical and non-critical switches in the Network. We defined the set of critical switches (SCS) as:

(9) SCS=∑i=1k⁡CSi

where k represents the Network’s total number of critical switches and gives us the number of subnets to partition the Network G. At the same time, we can obtain the set of non-critical switches from

(10) SnCS=S∖SCS

Therefore, by partitioning the OpenFlow switches S∈G into k sub-nets, namely, SDWAN_Partitionsi∀i=1,2,..,k according to the number of critical switches CS⊂V. The resulting SDWAN_Partitionsi can be defined as:

(11) SDWAN_Partitionsi=(Vi,Ei)

Such that:

(12) SDWAN_Partitionsiisacomponent

(13) ∑i=1k⁡CSi=1

(14) ∀i≠j∈k;SDWAN_Partitionsi∩SDWANPartitionsj={⌀}

(15) ⋃i=1k⁡Vi,⋃i=1k⁡Ei

Equation (12) indicates that the sub-net of any of the SDN_partitioni is made up of connected OpenFlow switches with links. Equation (13) ensures only one critical switch CSi is assigned to each partition. Equation (14) implies that an OpenFlow switches si can only be allocated to a single domain, while Eq. (15) ensures all the network switches are in one of the subnets. See Table 2 for the summary and description of symbols and notation used in our model.

Table 2 Notations and symbols.

Notation	Description	
G	SDWAN	
E	Set of communication links in the network	
V	Set of network nodes (comparison of both controllers and switches)	
C	Set of SDN controllers	
CProverhead	Controller overhead	
S	Set of OpenFlow switches	
SProverhead	Switch overhead on the controller	
CS	Critical switches	
nCS	Non-critical switches	
SCS	Set of critical switches	
SnCS	Set of non-critical switches	
SDWAN_Partitionsi	Sub-net of OpenFlow Switches	
dist(sicj)	Shortest distance between the controller cj and switch si in Sdomain	
k	An integer representing the number of CS, SDWAN_Partitions, and C	
nFi	New flow	
(nFSi,Sj)	Number of flow between source and destination	
Xim	{0,1} binary variables indicating whether the switch si is under the control of the controller Cm	

Network topology read phase

Algorithm 1 reads a GraphML file containing a network topology of SDWAN located at graphml_path. An empty graph object stores the network topology as G=(V,E) created in line 1 of the algorithm. V represents a set of switches in the Network, and E the physical communication links between the nodes. The network switch V comprised some OpenFlow switches S and SDN controllers’ C, i.e.,S,C∈V. However, the OpenFlow switches S consist of critical CS and non-critical switches nCS. The study defines a set of critical switches SCS in Eq. (9). Algorithm 1 reads the file to generate a graph object representing the network topology in line 2. Then, the algorithm returns the graph object in line 3 to identify these critical switches. The read_graphml function is a pre-existing function that reads and parses GraphML files.

Algorithm 1 ReadNetworkGraphTopology graph Cm.

Input: - graphml_path: the path to the GraphML file containing the network topology	
Output: - G: a graph object representing the network topology	
STAT of Algorithm	
 1. G ← new Graph()	
 2. G ← read_graphml(graphml_path)	
 3. For each si to sj∈G	
 4. Compute Nsp shortest path, Nsp(sisj)	
 5. Return, G,andNsp(sisj)	
END of Algorithm	

Switch role and critical switch identification phase (CSIP)

CSIP distinguishes between switches based on their roles to identify critical switches within a network. Because some switches within the network have a significantly higher frequency of communication with the SDN controller for rule installation than others. These switches are called critical switches because they impact the responsiveness of the SDN controller within the network. Therefore, a switch si∈Vi with high communication frequency with SDN controller for rule installation is considered more critical CsiI compared to an ordinary switch.

To establish the criticality of a switch si, we used the switch criticality metrics in a network, and the switch flow rule requests overhead on the controller. We assume that information in the network Gi from different sources si∀i=1,2..,N is propagated in parallel from the source si to the destination sj along the shortest path (geodesic), denoted as dij. Based on these assumptions, a switch si∀i=1,2..,N in a communication network Gi=(Vi,E) is critical to the extent of its criticality factor siCrf. Therefore, we use the switch’s connectivity in the network and its flow rule request overhead on the controller to model the switch criticality factor siCrf.

To determine the switch connectivity in the network, CSIP uses Algorithm 1 to return the number of shortest paths Nsp passing through the switch starting at si∈V and ending at sj∈V. Thus, we calculate the metric using the formula Eq. (16). On the other hand, to compute the switch traffic overhead on a controller, we consider the weighted new flow rule request sent from the source switch to the controller due to a new flow arrival based on Eq. (6) using Eq. (17). Following that, we compute the switch criticality factor siCrf using the formula presented in Eq. (18) using these parameters. Finally, we demonstrate the procedure for critical switch identification in Algorithm 2.

Algorithm 2 Critical switch identification.

Input: - G,andNsp(sisj):	
Output- {SCS,SnCS,CS_neighbours,distance}	
STAT of Algorithm	
 1. SCS← {}	
 2. SnCS← {}	
 3. FOR si∈V:	
 4.   siBC ← calculate switch connectivity in G using Eq. (16)	
 5.   sinFi ← calculate switch flow rule request using Eq. (17)	
 6.   siCrf ← calculate the switch criticality factor using Eq. (18)	
 7. total_siCrf ← sum_of_values (siCrf)	
 8. ave_siCrf ← total_siBC/length_of_values (siCrf)	
 9. FOR each si, in (siCrf):	
 10.  IF (siCrf) > ave_siCrf:	
 11.   add si and siCrf to SCS.	
 12.  ELSE:	
 13.   add siand siCrf to SnCS.	
 14. CS_neighbors ← {}	
 15. FOR each si, in siCrf:	
 16.  add a list of CS ’s neighbours to CS _neighbours.	
 17. distance ← {}	
 18. FOR each CS in SCS:	
 19.  For si, distance in shortest_path_length from CS in G:	
 20.   add ( si, CS) and distance to distance.	
 21. return SCS, SnCS, CS_neighbours, distance	
END of Algorithm	

(16) siBC=∑sisj∈V,si≠sjNsp(sisj|V)Nsp(sisj)

(17) sinFi=∑si∈S⁡(nFSi,cm)

(18) siCrf=siBC+sinFi

In (lines 1–2), Algorithm 2 initializes two empty dictionaries, SCSandSnCS. The dictionaries are used to store critical-switch and non-critical-switch information, respectively. For each switch si∈V in the SDWAN G, Algorithm 2 determines whether the switch si is critical or non-critical using Eq. (9) and by calculating its criticality factor (siCrf) using Eq. (18). The total (total_siCrf) and average (ave_siCrf) criticality factors for all switches in the network are also computed (lines 3–8). Algorithm 2 then checks the criticality factor (siCrf) of each switch si in the network topology G against the average criticality factor value (ave_siCrf) (lines 10–11). If (siCrf) is greater than (ave_siCrf), the switch is classified as critical and added to the set of critical_switch SCScontainers along with its criticality factor. Otherwise, it is classified as non-critical and added to the collection of non_critical_switch nSCS containers (lines 12–13).

Next, for each critical switch (CS) in the SCS container, Algorithm 2 retrieves the list of its neighbours and calculates its shortest path distance to all other switches in the network topology. The resulting information is added to the CS_neighbors and distances containers (lines 14–20). Finally, Algorithm 2 returns the sets of critical_switch, non_critical_switch, critical_switch_neighbors, and distances in (line 21).

Network partition based on switch criticality

The study designed a CSANP to partition the SDWAN (G) into smaller networks based on the number of critical switches ( num_CS). The CSANP collects inputs from Algorithm 2, where the critical switches of G are identified. The input parameters include the set of critical switches (SCS), non-critical switches (SnCS). The procedure is as shown in Algorithm 3. CSANP starts by initializing the number of Critical Switches (num_CS) and non-critical switches (num_nCS) on lines 1 and 2. It then calculates the average number of non-Critical Switches to be associated to each critcal switch and the remaining non-critical switches (num_CS_plus) on lines 3 and 4. The SDWAN_Partitions list is initialized with empty lists, where each list represents a partition associated with a critical switch (CS), on line 5. The algorithm then iterates through each non-critical switch (sj) in SnCS (line 6) and determines its closest critical switch (CS) based on the minimum distance (lines 7 to 14). The non-critical switch is then assigned to the corresponding partition in SDWAN_Partitions (line 14). Next, the algorithm iterates through each non-critical switch again (sj) (line 15) and assigns it to the appropriate partition in SDWAN_Partitions based on balancing criteria (lines 17 to 29). If a partition has fewer than avr_num_nCS, the current non-critical switch is added to it (line 24). If the partition has avr_num_nCS and there are remaining non-critical switches (num_CS_plus), one of them is added to the partition (lines 26 to 28). If the partition has avr_num_nCS, and there are no remaining non-critical switches, a new partition is created for the current non-critical switch (line 30). The process continues until all non-critical switches are assigned to partitions, and the resulting SDWAN_Partitions list contains the partitions, each associated with its respective critical switch. Finally, the algorithm returns the list of SDN [{SDWAN_Partitions},{SDWAN_Partitions}………|num_CS|]in line 31. Refer to the network partition formation phase of Fig. 2 for the flowchart for the algorithm.

Algorithm 3 Critical switch aware network partition (CSANP).

Input: (G, SCS, SnCS)	
Output: SDWAN_Partitions	
STAT of Algorithm	
1. num_CS = len(SCS)	
2. num_nCS = len(SnCS)	
3. avr_num_nCS = num_nCS // num_CS	
4. num_CS_plus = num_nCS % num_CS	
# Add all Critical Switches to SD-WAN partitions	
5. SDWAN_Partitions = [[] for _ in range(num_CS)]	
# Assign non-Critical Switch to Critical Switch based on minimum distance	
6. For sj in SnCS:	
7.  closest_CS = None	
8.  min_distance = float(‘inf’)	
9.  For i, si in enumerate(SCS):	
10.   dist = distance[si][sj]	
11.   If dist < min_distance:	
12.    min_distance = dist	
13.    closest_CS = i	
14.  SDWAN_Partitions[closest_CS] = SDWAN_Partitions[closest_CS] + [sj]	
   # Balance partitions and assign non-Critical Switches to Critical Switch	
15. For i, sj in enumerate(SnCS):	
16.  closest_CS = None	
17.  min_distance = float(‘inf’)	
18.  For j, si in enumerate(SCS):	
19.   dist = distance[si][sj]	
20.   If dist < min_distance:	
21.    min_distance = dist	
22.    closest_CS = j	
23.  cluster_index = closest_CS	
24.  If len(SDWAN_Partitions[cluster_index]) < avr_num_nCS:	
25.   SDWAN_Partitions[cluster_index] = SDWAN_Partitions[cluster_index] + [sj]	
26.  Elif len(SDWAN_Partitions[cluster_index]) < avr_num_nCS + 1 and num_CS_plus > 0:	
27.   SDWAN_Partitions[cluster_index] = SDWAN_Partitions[cluster_index] + [sj]	
28.  num_CS_plus -= 1	
29.  Else:	
   # If no condition is met, create a new partition for the non-Critical switches	
30.   SDWAN_Partitions = SDWAN_Partitions + [[sj]]	
31. return SDWAN_Partitions	
END of Algorithm	

Critical switch aware controller placement (CSACP)

The proposed Critical Switch Aware Controller Placement (CSACP) algorithm is responsible for placing an SDN controller in each of the resulting network partitions (subnets) produced by CSANP. This placement problem is a variant of a facility location problem. Therefore, for each of the resulting subnets [{SDWAN_Partitions1},…{SDWAN_Partitions|num_CS|}] obtained from the CSANP, we designed a CSACP algorithm to place the SDN controller on each SDWAN_Partitionsi=(Vi,Ei) within the shortest distance of each demand point in the subnets. We assigned C to represent the set of controllers cj∈C∀j=1,2…,m for the k sub-nets. Next, for each, ∀SDWAN_Partitionsi, our placement model maps the controller cj∈C∀j=1,2…,m to the demand points si∈V, which are the OpenFlow switches, in a way that the dist(sicj) is the shortest distance between the candidate controller locations j∈SDWAN_Partitionsi and the mapped controller cj∈C. Thus, the proposed CSACP algorithm finds a suitable position in each resulting partition to place the controller. Algorithm 4 provides a detailed description of the proposed controller placement method.

Algorithm 4 Critical switch aware controller placement (CSACP).

Input: {SCS,SnCS} [{SDWAN_Partitions1},…{SDWAN_Partitions|num_CS|}]	
Output- controller_positions	
STAT of Algorithm	
 1. controller_positions = {}	
 2. For SDWAN_Partitions_num, partition in enumerate(SDWAN_Partitions) Do	
 3.  max_critical_switch = null	
 4.  max_siCrf = -1	
 5.  For switch in partition, Do	
 6.   If switch in critical_switch and critical_switch[switch] > max_siCrf Then	
 7.    max_critical_switch = switch	
 8.    max_siCrf = critical_switch[switch]	
 9.   End If	
 10.  End For	
 11. distances_within_partition = {}	
 12. For a node in partition, Do	
 13.  If the node in non_critical_switch, Then	
 14.   distances_within_partition[node] = distances[(node, max_critical_switch)]	
 15.  End If	
 16. End For	
 17. min_distance_node = null	
 18. min_distance = infinity	
 19. For a node in distances_within_partition, Do	
 20.  If distances_within_partition[node] < min_distance, Then	
 21.   min_distance_node = node	
 22.   min_distance = distances_within_partition[node]	
 23.  End If	
 24. End For	
 25. controller_positions[SDWAN_Partitions_num] = (max_critical_switch, min_distance_node)	
 26. End For	
 27. return controller_positions.	
END of Algorithm	

(19) Min1|SDWAN_Partitionsi|∑si∈SDWAN_Partitionsi⁡dist(sicj)

Such that

(20) si,cj∈SDWANPartitionsi

The proposed CSACP algorithm takes inputs from CSANP (Algorithm 2), which includes the SDWAN partitions, critical and non-critical switches, and their criticality factors. Each partition is a set of switches within the SDWAN network. The algorithm initializes an empty dictionary called controller_positions to store the controller positions for each SDWAN partition in line 1. Then, for each partition in the input set of partitions, the algorithm identifies the critical switch with the highest criticality factor max_siCrf. In (lines 2–11), Algorithm 4 calculates the distance to the identified critical switch using a pre-computed distance metric stored in a distance dictionary for each non-critical switch in the partition. Next, the algorithm finds the non-critical switch within the partition that has the minimum distance to the identified critical switch and assigns it as the controller position for that partition. The algorithm then stores the controller position for that partition in the controller_positions dictionary in (lines 12–26). Finally, the algorithm returns the controller_positions dictionary as the algorithm output in line 27.

Experimentation setup and performance evaluation of CPCSA

In this section, the performance of CPCSA is evaluated and compared with other representative solutions in the literature. The study utilizes three (3) real network topologies obtained from the Internet Topology Zoo (ITZ) (A. G. University of Adelaide, 2023) and randomly generates topologies for conducting the experiments. The database provides researchers access to hundreds of real network topologies from various service providers. Thus, the study selects AsnetAm, Arpanet19728, and ARNES networks for the experiments. Table 3 gives additional information on other aspects of the chosen network topologies, which vary in size and structure. The partitioning phase is performed offline with a script written in Python 3.8.0 and NetworkX components. The experiment uses Mininet version 2.3.0 to build the topologies of these partitions with an OpenvSwitch for interaction with a Ryu SDN controller in each partition based on OpenFlow v1.5.1 specifications. The article borrows traffic matrix scenarios in the GÉANT network (Uhlig et al., 2006) for understanding traffic patterns. The traffic matrix of (Uhlig et al., 2006) describes the traffic between nodes and its transfer speed, highlighting what constitutes a new flow. A D-ITG utility injects a TCP/UDP flow on 1,024 Mbps transmission lines of the Mininet architecture to generate the traffic. Hence, the study model, one new flow for every 100,000 KB, exchanged, according to Poisson traffic distribution in terms of Packet Inter Departure Time (PIDT). The reliance of the packet_IN message on whether the switch piggybacked the first packet of a flow to a controller (Yusuf et al., 2023c). The article considers its size and Packet count as in Obadia et al. (2015) to account for it. Additionally, as proved in Obadia et al. (2015), there must be a packet OUT message (flow_mod Packet) for every packetIN message; thus, the study considers their sizes and packet count equal.

Table 3 Topologies information and traffic information.

Topologies information	Traffic information	
Topology	Number switches	Number of links	Density	Ave SBF	New flow	Packet_IN msg size	Packet_OUT msg size	
Arpanet19728	29	32	0.0788	0.136	For every 100,000 Kb	80 bytes	80 bytes	
ARNES	34	47	0.0837	0.076	
AsnetAm	65	79	0.0380	0.044		

We start off the evaluation of CPCSA by providing a visual representation of its controller placement result in Fig. 3. We then presented the overhead incurred by the controller placed in a network using the proposed CPCSA compared to other related CPP solutions in Fig. 4. While in Fig. 5, the study investigates the impact of CPCSA on fault tolerance by evaluating the rate of control packet loss. Lastly, the evaluation of Throughput and average switch-to-controller Latency is done in Figs. 6 and 7, respectively. We conduct all the experiments on a machine with Intel(R) Core (TM) i7-10750H CPU @ 2.60 GHz, 2.59 GHz, and 16.0 GB memory.

Figure 3 (A–D) Arpanet topology; (E–H) Arnes topology; (I–L) AsnetAm topology.

Figure 4 (A–D) Overhead.

Effect of flows installation cost on the overhead on the number of controllers.

Figure 5 (A and B) Packet loss result.

Comparison of packet loss.

Figure 6 (A and B) Throughput.

Comparison of throughput.

Figure 7 (A and B) Latency.

Relationship between switch to controller latency.

Results

Network partitions and controller placement positions

The diagrams presented in Figs. 3A–3I illustrate the network partitions and selected positions for controller placement as determined by the proposed CPCSA algorithm. Figure 3 depicts the outcomes of the controller placement output when applied to the Arpanet19728, ARNES, and AsnetAm topologies. As demonstrated in Figs. 3A, 3E, and 3I, before network partitioning, node 4, node 7, and node 22 are designated as the controller positions. This selection occurs based on the switch criticality factors siCrf ranging from 0.25, 0.50–0.61, to 0.59–0.66 in the respective topologies. Conversely, as shown in Figs. 3B, 3F and 3J, when the switch criticality factors are 0.25, 0.18–0.49, and 0.27–0.55 in the corresponding networks, the networks are partitioned into two subnets. Consequently, in Arpanet19728, nodes 4 and 13 are chosen as the controller positions, while in ARNES, nodes 7 and 30 are selected. In the AsnetAM topology, the controller positions are nodes 22 and 7. Furthermore, by reducing the switch criticality factors siCrf to 0.22, 0.14–0.15, and 0.15–0.25, the respective networks experienced partitioning into four subnets. This resulted in the inclusion of nodes 23 and 28 as additional controller positions in the Arpanet19728 topology. Similarly, in the case of ARNES, nodes 23 and 29 were selected as new placements, while for AsnetAM topology, CPCSA chooses nodes 8 and 26 to place the new controllers. Please refer to Figs. 3D, 3H, and 3L for visualization

Controller overhead

Figure 4 shows the accumulated controller’s rule installation overhead in the Arpanet19728, ARNES, and AsnetAm network topologies with SPDA (Guo et al., 2022), gravCPA (Wang, Ni & Liu, 2022), and the proposed CPCSA, respectively. The experiment results show that CPCSA incurred lower rule installation overhead than SPDA (Guo et al., 2022) and gravCPA (Wang, Ni & Liu, 2022) in all the topologies. As shown in Fig. 4A, the proposed CPCSA had reduced the SDN controller’s overhead compared to SPDA and gravCPA in the AsnetAM topology by 63% and 49%, respectively. Meanwhile, in Fig. 4B, with the Arnes topology, the proposed technique is shown to cut the overhead by 54% and 36%. Lastly, CPCSA minimizes the overhead of SPDA (Guo et al., 2022) and gravCPA (Wang, Ni & Liu, 2022) by 63% and 51% in the Arpanet19728 topology, as revealed in Fig. 4C. The achievement of the overhead reduction is attributable to the control of the number of critical switches CPCSA assigns to a single SDN controller. A switch is critical if it continually appears along the shortest path of many dissimilar host-to-destination communicating pairs. This type of switch receives an augmented number of rule installation instructions from the controller on what to do with the flow. Because, by default, flows are usually routed along the shortest path from the source to the destination host in most networks. Thus, the controller with a higher number of critical switches in a partitioned SDWAN incurs higher overhead. The additional controller overhead will amount to the number of switches assigned to the controllers by a factor of their generated control traffic.

Control packet loss

In this section, this study measures the impact of control packet loss during switch-to-controller communication to verify CPCSA’s fault-tolerance benefits. High control plane overhead can induce a network problem, which can cause some switches to lose connections with their controllers, resulting in dropped packets. The study expects CPCSA to reduce the possibility of Network failures owing to excessive controller overhead, which can lead to substantial packet loss. Because, by design, the CPCSA differentiates among network switches and restricts the number of critical switches for each partition. We use Python 3.8.0 with NetworkX and Matplotlib library components for simulation. However, unlike the previous experiments with real network topologies, fully connected networks are randomly generated using Barabási–Albert (BA) model. After 50 repeated experiments, the average results findings in comparison to alternative approaches are shown in Fig. 5. The y and x-axis in Fig. 5 display the average control packet loss as a function of the x-axis representation of the total network nodes, n. As expected, CPCSA has the lowest average packet loss rate of the four routing algorithms due to minimising the controller’s overhead. On DBCB, the proposed CPCSA reduced packet loss by 31%, while on SPDA and gravCPA, it reduced it by 61%. The minimum controller’s overhead correlates better with preventing network failure and lower control packet loss. Therefore, a low average control packet loss indicates the technique’s ability to avoid network faults due to high overhead.

Throughput

Figure 6 displays the network throughput evaluation result between the proposed CPCSA and the benchmark algorithms. The Throughput metric gives information about the performance of the techniques regarding the number of control data packets sent from a source host and successfully delivered at the destination host during a transmission period (Guo et al., 2022). The throughput metric is relevant in assessing CPCSA performance about how it reacts to network-changing events that can trigger flow setup requests or failure. Figure 6A shows the result of CPCSA’s throughput with different numbers of controllers. Figure 6B shows the CPCSA’s Throughput vs that of gravCPA (Ali, Lee & Roh, 2019) and SPDA (Obadia et al., 2015). As can be seen from Fig. 6B, CPCSA outperformed the benchmarked reference algorithms. Comparatively, the algorithm improved the throughput achieved by gravCPA and SPDA by 16% and 18%, respectively. This improvement indicates that the methodology adopted by CPCSA to minimise the controller’s overhead significantly influenced the control packet delivery rate. Thus, this analysis affirms the research question: “Can controlling the number of critical switches under the control of an SDN controller improve the Quality of Service in a network?”

Switch to controller average latency

In this subsection, the study demonstrates how the average switch-controller latencies respond when a controller is appropriately placed in the subnets of the network partitioned while considering critical switches. For validation and revelation of results, the study compares the performance of CPCSA with that of other controller placement solutions that incorporate a network partitioning strategy and allocation of a controller to each subnetwork. In the experiments, we ensure that all the benchmarked algorithms deploy the same number of controllers as CPCSA in the network for a fair evaluation. Therefore, given a controller cj∈C and the switches si∈SDWAN_Partitionsi in the sub-network, the CPCSA uses the relation in Eq. (17) to measure the latency metrics. Based on the result obtained, Fig. 7 displays the relationships between the average switch-controller latencies with the number of controllers and partitions varying from 1 to 4 on three (3) topologies. As shown in Fig. 7, the result exhibits a monotonic decreasing trend in the switch-controller Latency with an increasing number of partitions and controllers. We observed this pattern throughout all four (4) algorithms under study. i.e., Increasing the number of controllers and partitions causes all the compared algorithms to behave identically regarding average switch-controller control packet processing delay. However, CPCSA performs significantly better when compared to SPDA, DBCP, and gravCPA algorithms. As shown in Fig. 7A, the proposed CPCSA reduces the average switch-to-controller Latency by 27%, 12%, and 3%, respectively, compared to SPDA (Guo et al., 2022), DBCP (Liao et al., 2017), and gravCPA (Wang, Ni & Liu, 2022) algorithms when the Algorithms partitioned the network into 4.

Conclusions

The controller placement algorithm with network partition based on critical switch awareness (CPCSA) is a novel approach to address the challenge of transient congestion due to controllers’ overhead in the existing controller placement problems (CPP) solutions in SDN. CPCSA identifies the set of critical switches in a network to guide the network partition procedure for finding the optimal number of controllers and placement in the network. The algorithm has been implemented and evaluated in a laboratory testbed in a series of comparative experiments with similar solutions using multiple Real life network topologies from ITZ. The comparative experiments demonstrate CPCSA’s effectiveness in reducing control message overhead, control packet loss, switch-to-controller latency, and improved throughput. The results show that the proposed solution has achieved an aggregate reduction in the controller’s overhead by 73%, loss by 51%, and latency by 16% while improving throughput by 16% compared to the benchmark algorithms. However, the proposed scheme does not support heterogeneous controllers and has no defense mechanism against vulnerabilities such as DDOS, common-mode fault, etc.

For future research, we plan to update the CPCSA controller placement model with traffic flow behavioural quality of service requirements for consideration. It would be intriguing to employ machine learning techniques such as deep learning to study flow behaviour based on flow history for the classification. Considering this would support designing a controller placement with traffic dynamics awareness. The aim is to partition the network and place a controller while considering the traffic pattern in the network. Another exploration avenue could be integrating the algorithm with heterogeneous controllers’ support. We can see the motivation for these from many perspectives. First, a homogeneous CP provides a potential security risk due to the controllers’ common-mode fault, often known as a common vulnerability point. Assume enemies are aware of the vulnerability of one controller; in this instance, they can easily knock down the entire network by exploiting the controller’s shared vulnerability. Second, interoperability between various controller platforms and traditional IP networks can encourage and facilitate the commercial adoption of SDN globally. Very little research has examined this direction thus far. Therefore, undertaking further research in this direction will be a valuable contribution.

Supplemental Information

Supplemental Information 1 CPCSA Main Code.

Source code of the main file that import the CPCSA as a module.

Click here for additional data file.

Supplemental Information 2 CPCSA code.

Click here for additional data file.

Supplemental Information 3 Raw Data results.

Click here for additional data file.

Additional Information and Declarations

Competing Interests

Author Contributions

Data Availability

The authors declare that they have no competing interests.

Nura Muhammed Yusuf conceived and designed the experiments, performed the experiments, performed the computation work, authored or reviewed drafts of the article, and approved the final draft.

Kamalrulnizam Abu Bakar conceived and designed the experiments, analyzed the data, prepared figures and/or tables, authored or reviewed drafts of the article, and approved the final draft.

Babangida Isyaku performed the experiments, analyzed the data, performed the computation work, authored or reviewed drafts of the article, and approved the final draft.

Abdelzahir Abdelmaboud performed the computation work, prepared figures and/or tables, and approved the final draft.

Wamda Nagmeldin performed the computation work, prepared figures and/or tables, authored or reviewed drafts of the article, and approved the final draft.

The following information was supplied regarding data availability:

The raw data and code are available in the Supplemental Files.

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
