# Peer review of "Controller placement with critical switch aware in software-defined network (CPCSA)"

_PeerJ Computer Science, doi:10.7717/peerj-cs.1698_

## Round 0.1 · original submission · Minor Revisions

Although enriching with more recent and related references is needed, It is not essential to have specific ones mentioned in the reviewer comments.
More clarification for the contribution in the abstract is highly recommended.

Reviewer 1 ·

Basic reporting

clear
good structure
clear results

Experimental design

orignal
primary research
described well

Validity of the findings

conclusions supported by results

Additional comments

In this paper the authors have proposed a controller placement problem approach leveraging the critical switch awareness in SDN (software defined networking).
I recommend the following changes (must) be addressed by the authors:
Enlist the limitations of each work in the literature review.
Moreover, include some recent literature i.e. on the controller placement problem in SDN such as:
Ali, J. and Roh, B.H., 2022. An Effective Approach for Controller Placement in Software-Defined Internet-of-Things (SD-IoT). Sensors, 22(8), p.2992. and
Ali, J., Lee, S. and Roh, B.H., 2019, June. Using the analytical network process for controller placement in software defined networks (poster). In Proceedings of the 17th Annual International Conference on Mobile Systems, Applications, and Services (pp. 545-546).
Describe the motivation for the controller placement and novel contributions of your works.
What is the controller overhead? Can the authors represent it mathematically with the help of an equation or formula.
Include some future research directions.

·

Basic reporting

Please follow my comments to improve the manuscript.

1. Write the full meaning of SDWAN in abstract.

2. In "Algorithm2: Critical Switch Identification" and Equation no 17, what is SiBC? Discuss in detail.

3. Include a Flowchart for "Algorithm 3: Critical Switch Aware Network Partition (CSANP)"

4. Include more citations from latest research paper.

5. Write a major weak point of the research in conclusion.

Experimental design

In experimental design include the processing time to Placement.

Validity of the findings

Good research work. Discuss detail about How many typology grabbed from Internet Topology Zoo?

Additional comments

Correct the position of Equation number and improve the English.

---

## Round 0.2 · accepted · Accept

Thanks for addressing the reviewer's comments properly.

Reviewer 1 ·

Basic reporting

Clear now

Experimental design

methods are described with sufficient details and relevant information

Validity of the findings

conclusions are well stated

Additional comments

The authors have addressed my comments well.

·

Basic reporting

Thank you for following my comments to improve the manuscript.

Experimental design

Good work

Validity of the findings

Good research